# OpenReview forum: "Walk the Talk? Measuring the Faithfulness of Large Language Model Explanations"
_ICLR.cc/2025/Conference — ICLR 2025 Spotlight_

### Official Review · Reviewer_5TDz · 2024-10-30

**Soundness:** 3
**Presentation:** 4
**Contribution:** 3
**Rating:** 8
**Confidence:** 4

**Summary:**

The paper investigates ‘unfaithfulness’ of SOTA LLMs. In this paper, the authors introduce a new metric for measuring the faithfulness of a model, namely, causal concept faithfulness that not only quantifies but also reveals semantic patterns of unfaithfulness. To uncover these patterns, they put this method to the test on two tasks - a social bias task and a medical QA task to demonstrate how decisions made by the models change along with the provided explanations to justify the wrong decisions made by the model.

**Strengths:**

I really enjoyed reading the paper! It addresses a highly relevant topic of faithfulness in the current landscape of AI. The writing is engaging and clear.

One of its standout features is its approach to a critical issue: it offers a concrete and measurable method for assessing explanation faithfulness in large language models, an area that has been difficult to define in previous research. By introducing the concept of causal concept faithfulness, the authors provide a way to evaluate how "honest" a model's explanations are, while also revealing specific patterns of misleading explanations.

**Weaknesses:**

**High level comments**
1. It remains uncertain whether biases impact all types of reasoning tasks uniformly or if certain domains are more affected than others.
2. Moreover, the experiments do not specify how these findings may apply beyond classification tasks to biases that could affect other generative tasks.

***Minor comment***
Line 321 typo -> should be orange for behavior, red for identity

**Questions:**

1. Can a faithful explanation be biased? For example, suppose if both CE and EE are high for the example in Table 1, and if the model would have answered Male: 26% Female: 74%, with Explanation References: Traits/Skills: 15% Age: 0% Gender: 85%. This is a clear case of gender bias, but can we say the explanation is faithful here referring to Definition 2.3?
2. How can we understand the observations if the models memorized these benchmark datasets during their training stage? Does this imply that the model’s reasoning process is so vulnerable to bias that it can even disregard a previously memorized correct answer?

---

> ### Author Response · Authors · 2024-11-22
>
> We thank the reviewer for their careful review of our work and their thoughtful feedback. We appreciate that they recognize that our work tackles a critical issue and that it provides a concrete solution to a difficult problem. We are grateful that the reviewer found our paper to be engaging and enjoyable to read.
>
> We address each comment below:
>
> **It remains uncertain whether biases impact all types of reasoning tasks uniformly or if certain domains are more affected than others.**
>
> Although we did not investigate all types of reasoning tasks, we did choose to analyze two tasks that are substantially different from each other. The first, a social bias task, contains subjective questions that are designed to elicit stereotype-based reasoning. The second, a medical question answering task, contains objective questions that are intended to be answered with logical/fact-based reasoning. Whereas one might expect LLMs to produce unfaithful explanations on the first task, it is less clear what to expect on the second task. We think a notable contribution of our work is demonstrating that LLMs can produce unfaithful explanations on tasks with such stark differences. We see this as early evidence that unfaithfulness might be common across many different tasks, as opposed to just those that are clearly prone to bias. We think the recognition of this risk is important knowledge for the AI community – and it is thanks to our method that we are aware of it. In future work, we would like to investigate the prevalence of faithfulness across tasks more extensively by applying our method to additional domains.
>
> **Minor comment Line 321 typo -> should be orange for behavior, red for identity**
>
> Thank you for catching this. We have fixed it in the paper.
>
> **Q: Can a faithful explanation be biased? For example, suppose if both CE and EE are high for the example in Table 1, and if the model would have answered Male: 26% Female: 74%, with Explanation References: Traits/Skills: 15% Age: 0% Gender: 85%. This is a clear case of gender bias, but can we say the explanation is faithful here referring to Definition 2.3?**
>
> Good question. Yes, LLMs can be influenced by bias yet still produce faithful explanations. Your example is a nice illustration of that point – if an LLM’s decisions are influenced by gender bias and its explanations admit this bias, then we consider the explanations to be faithful. This is captured by Definition 2.3: if both the CE and EE are high for gender compared to the other concepts, then this contributes to a high faithfulness score.
>
> **How can we understand the observations if the models memorized these benchmark datasets during their training stage? Does this imply that the model’s reasoning process is so vulnerable to bias that it can even disregard a previously memorized correct answer?**
>
> We are not sure if the LLMs we study have seen the datasets we use in our experiments during training, but it is a possibility. As you allude to in your comment, in our experiments on the social bias task, we find that there are cases in which the LLM doesn’t provide the correct answer and instead selects the bias-aligned answer. If an LLM had seen these questions during training, then it seems like there are at least two reasons why the model might not consistently select the correct answer: (1) despite seeing these questions, it didn’t memorize them or (2) it has memorized them, but because of differences in the prompt we use vs the prompt used during training, our prompt doesn’t trigger the memorized response.

---

### Official Review · Reviewer_uDtA · 2024-11-03

**Soundness:** 3
**Presentation:** 3
**Contribution:** 3
**Rating:** 8
**Confidence:** 3

**Summary:**

A formulation of faithfulness is presented called causal concept faithfulness. According to this formulation, when a model produces a natural language explanation of its behaviour that appeals to certain concepts (e.g. gender), altering the input to have a different concept value (e.g. changing the gender of person mentioned in the text), should also alter the behaviour. Thus, under this formulation, a model is faithful if and only if it appeals to exactly those concepts that—if altered—would actually change its behaviour. To measure faithfulness the correlation between two metrics is used: (1) the probability that a concept is mentioned in an explanation; and (2) the actual change after altering the inputs, the KL divergence between the model's output distribution before and after alteration is used.
To avoid having to measure these values on very large datasets, the authors propose to use a Bayesian hierarchical model, which 'partially pools' information across interventions for related concepts.
Experiments are performed on two tasks. The first is a task designed to elicit unfaithful explanations. Models perform poorly w.r.t. two out of three concepts. In the second task, models are shown to have limited faithfulness when explaining their answers for medical question answering.

**Strengths:**

- Precise definition of what is meant by faithfulness.
- 'causal concept faithfulness' as proposed will be useful for the explainability community.
- The paper is written well, while being information-dense.

**Weaknesses:**

- The use of GPT4o to generate counterfactual inputs is not evaluated independently.

**Questions:**

- In Figure 1, it appears as though the correlation between EE and CE would be significantly lower if done independently for each concept, and then averaged. My question is: is calculating faithfulness on a per-concept basis is possible with your method?
- And a related question, given that pearson correlation only measures to what extent points lie on *a* line, and not on *the* line y=x, is it the most appropriate metric for you use case, did you consider others?

---

> ### Author Response · Authors · 2024-11-22
>
> We thank the reviewer for their careful review and for providing useful feedback and questions. We appreciate their positive comments regarding our precise definition of causal concept faithfulness and its utility for the explainability community.
>
> We address each comment below:
>
> **The use of GPT4o to generate counterfactual inputs is not evaluated independently.**
>
> We conducted a careful manual review of a large random sample of the counterfactual questions to validate their correctness. For each question, we checked that: (1) the requested edit was made and (2) no other information in the question was altered. For BBQ, we reviewed all counterfactuals (268 total). For MedQA, we reviewed the counterfactuals for a random sample of 15 out of the 30 questions (161 counterfactuals total). We have now included a random sample of counterfactuals in Appendix D.2. For BBQ, the removal-based counterfactuals are in Table 19 and the replacement-based counterfactuals are in Table 20. For MedQA, all counterfactuals are removal-based and they are in Tables 22 and 23.
>
> In addition, we would like to point out that GPT-based models have been used in prior work for counterfactual generation and have been found to produce high-quality counterfactuals [1][2].
>
> [1] Gat, Yair, et al. "Faithful explanations of black-box nlp models using llm-generated counterfactuals." arXiv preprint arXiv:2310.00603 (2023).
> [2] Polyjuice: Generating Counterfactuals for Explaining, Evaluating, and Improving Models (Wu et al., ACL-IJCNLP 2021)
>
> **Q: In Figure 1, it appears as though the correlation between EE and CE would be significantly lower if done independently for each concept, and then averaged. My question is: is calculating faithfulness on a per-concept basis is possible with your method?**
>
> This is an interesting observation. When you say “concept,” we’re assuming you are referring to the concept categories (i.e., behavior, context, and identity) / colors in the plot. Please correct us if that is not the case. It is possible to calculate faithfulness on a per-concept category basis with our method. This can be done by computing the correlation for a subset of the concepts (e.g., just the “behavior” concepts) at a time. We think this would be an interesting analysis to explore in future work.
>
> **And a related question, given that pearson correlation only measures to what extent points lie on a line, and not on the line y=x, is it the most appropriate metric for you use case, did you consider others?**
>
> This is a good question. Though Pearson correlation is a commonly used metric, as you point out, there are other options. We chose Pearson correlation because it is simple and easily understood. We discuss the other metrics we considered below:
> 1. *Error metrics, such as mean squared error (MSE) and root mean squared error (RMSE)*. We decided not to use these metrics because the two scores that we compare have different scales: causal concept effects range from 0 to infinity (as they are based on KL divergence), whereas explanation implied effects range from 0 to 1. Consequently, we do not expect these two scores to have the exactly same values, which limits the applicability of these metrics.
> 2. *Rank Correlation Metrics (e.g., Spearman’s rho, Kendall’s tau)*. These metrics assess the similarity of the orderings of concepts when ranked by the two scores. One limitation is that they penalize all misrankings equally, regardless of the values of misranked items. This means that misranking two concepts with very similar concept effects receives the same penalty as misranking two concepts with a large difference in concept effects. In preliminary experiments using Kendall’s tau as the alignment metric, we found that this sometimes led to unintuitive results.

---

### Official Review · Reviewer_7q6M · 2024-11-03

**Soundness:** 3
**Presentation:** 3
**Contribution:** 3
**Rating:** 8
**Confidence:** 4

**Summary:**

**Summary**:
This paper adopts a causal inference approach to define and evaluate the faithfulness of LLM-generated explanation in the context of two question answering tasks. The obtained results X.

**Main contributions**: The main contributions are the definition and methodology proposed to assess the faithfulness of explanations.

**Methodology**:
- Key to the methodology is the assumption that a model is faithful if its explanations consist of only concepts that are impactful for the decision (i.e., have large causal effects) (lines 183-185).
- The authors first compute the causal effect associated with each concept in the explainability (CE). An auxiliary LLM is used to determine the _explainable_ concepts and produce counterfactual perturbations for each input x. CE is then estimated by contrasting the distributional differences between LLM responses when given the modified inputs vs the original inputs.
- Then the authors determine the prevalence of each concept appearing in the explanation (EE).
- Finally, the authors determine the linear alignment between CE and EE (dubbed causal concept faithfulness) for each example using the pearson correlation coefficient. Dataset-level faithfulness score is the average over the examples.

**Writing**: Overall the writing is clear and easy to follow! The authors did a good job in exposing the ideas. Consider the following comments to further increase clarity:
- Add information about when the experiments were run with each model.
- lines 321-323: you describe the colors for each of the concept categories. However there seems to be a mismatch between the category color in the image and the color described in text.

**Strengths:**

-  Causally inspired definition and methodology to assess the faithfulness of LLM-generated explanations.
- Empirical validation of proposed methodology in two different question answering datasets.
- The finding that GPT-3.5 produces more faithful explanations (at a dataset-level) than the more recent and advanced models (GPT-4o and Claude 3.5 sonnet) is interesting. They also show that unfaithful explanations by GPT-3.5 is more harmful than GPT-4o
- The analysis concerning the impact of different types of interventions (i.e., remove concept vs swap it with a different value) is interesting, revealing the brittleness of safety guards employed in GPT-3.5 and GPT-4o.

**Weaknesses:**

1. **Insufficient validation of the proposed approach**: the authors mention that their method can be used to quantify and discover interpretable patterns of unfaithfulness. However, there is no guarantee that the methodology detects truly unfaithful concepts. To further ensure the correctness of the approach, it would be nice to show linear agreement between CE and EE in a controlled setting where LLMs are known to only produce faithful explanations (e.g., unbiased setting).
2.  **Important parameters of the experiments are not clear in the paper, which may affect reproducibility of the experiments**. The authors could consider providing additional details about the exact number of perturbations generated for each example, the number of generations used to estimate P(Y|X) (during the definition of CE), the decoding hyperparameters (including the temperature and max number of tokens). Additional details should also be provided about how each response y is parsed – this is particularly relevant given that the evaluated models are known to produce nuanced and semantically equivalent texts.
2. **Small evaluation set and concerns about generalizability**: for the two question-answering, the authors estimate the causal effects (CE) and explanation-implied (EE) effects for 30 examples (using 50 generations per each dataset). However, it is unclear how robust these results are and whether these would generalize to larger models.  Perhaps the authors could show how the faithfulness score varies as the number of datapoints increases, therefore, providing an idea of the variability and stability of the scores.
3. **Univariate counterfactuals**: if I understood correctly, the proposed framework focuses on perturbing the sentences one concept at a time, irrespective of the correlations between features. However, this fails to account for the correlations between different features (e.g., name of schools or organizations is related to socio-demographic features).
4. **Generalizability to open-source models**: the paper carries analyses on two OpenAI models (GPT-3.5, GPT-4o) and one Anthropic model (Claude-3.5-sonnet). Could be interesting to contextualize the faithfulness of existing open source models (e.g., Llama 3.1) and draw comparisons with closed-source ones.

**Questions:**

1. One of the decisions in the paper is to use an auxiliary model to extract concepts, propose a list of alternative values for each concept. Why is this necessary and is there any assumption or requirement that helped settling for a GPT-4o as the auxiliary LLM? The authors could better motivate the selection of GPT-4o in their paper, perhaps by including a small human study comparing the effectiveness of different models in extracting concepts and creating the list of alternate values. The authors should also consider including the prompt used to extract concepts and concept values in the Appendix.
2. In line 218, the authors mention the use of auxiliary LLM to “list distinct concepts in the context of x”. What kind of verifications were performed to ensure that the extracted concepts and their list of concepts were meaningful? The authors should consider adding a list of extracted concepts and their list of values to the Appendix. They should also consider adding more details about the validation (e.g., manual validation or llm-as-a-judge approach).
3. Similarly, to the two questions above, in line 224-225, the authors mention “to generate each counterfactual, we instruct [auxiliary LLM] to edit the question x by changing the value of [concept] ..., while keeping everything else the same”. However there seems to be no validation of this claim. Did the authors validate that the perturbed input x was indeed minimally distant from x? If not, the authors should consider including such analysis, perhaps by showing the minimum edit distance or n gram overlap between the modified and original inputs.
4. Why did the authors select a linear correlation coefficient as opposed to a non-linear coefficient?
5. In Figure 1 (right), we observe that different behavioral concepts end in different regions of the scatter plot (there are orange points in the top and orange points around EE in [-0.5, -1.5]. Is there any insight or pattern that justify why there are different clusters? Could it be that the model is less prone to use some concepts for specific demographics?

---

> ### Author Response · Authors · 2024-11-22
>
> We thank the reviewer for their thorough review. We appreciate their insightful feedback and thoughtful comments and questions, which have helped us to strengthen the paper. We have now uploaded a new version with changes based on their feedback.
>
> We have grouped the reviewer’s comments by theme and addressed each below.

---

> > ### Author Response · Authors · 2024-11-22
> > **Additional experiments to improve the validation of our method**
> >
> > Thanks to the reviewer’s feedback, we now have added new experiments to improve our experimental validation: (1) we analyze our method in a faithful setting, (2) we examine the robustness of our method to dataset size, and (3) we are in-progress on experiments with opensource LLMs, and will provide an update on that before the end of the discussion period.
> >
> > **Insufficient validation of the proposed approach: the authors mention that their method can be used to quantify and discover interpretable patterns of unfaithfulness. However, there is no guarantee that the methodology detects truly unfaithful concepts. To further ensure the correctness of the approach, it would be nice to show linear agreement between CE and EE in a controlled setting where LLMs are known to only produce faithful explanations (e.g., unbiased setting).**
> >
> > It sounds like the reviewer is concerned with the potential for our method to report that LLM explanations are unfaithful, when they are actually faithful. We agree that this is an important concern. To address it, we have added an analysis focused on the setting in which we expect LLMs to produce faithful explanations, as you’ve suggested. In the updated version of the paper, we refer to this analysis in Section 4.1 in the main body (lines 415-417) and include the details in Appendix E.2. In this new experiment, we focused on the small subset of questions in the BBQ dataset that have objective answers (e.g., the top question in Table 28 says that one individual “stayed in the driver’s seat” and then asks “who stayed in the car?”). When answering these questions, we expect LLMs to use the evidence (since it is conclusive) rather than relying on stereotypes, and hence, we expect LLM explanations to be faithful. Our results confirm that we find this expected trend. On the objective BBQ questions, all models obtain high faithfulness scores of $\mathcal{F}(\mathbf{X}) \geq 0.95$ (where  $\mathcal{F}(\mathbf{X}) = 1$ is perfectly faithful). See Appendix E.2 for further details.
> >
> > **Small evaluation set and concerns about generalizability: for the two question-answering, the authors estimate the causal effects (CE) and explanation-implied (EE) effects for 30 examples (using 50 generations per each dataset). However, it is unclear how robust these results are and whether these would generalize to larger models. Perhaps the authors could show how the faithfulness score varies as the number of datapoints increases, therefore, providing an idea of the variability and stability of the scores.**
> >
> > We thank the reviewer for pointing out this valid concern. As suggested, we have now added an analysis of the robustness of our method to the dataset size, which shows that the results are stable.  Specifically, we repeated our analysis with the number of examples N = 5, 10, 15, 20, 25, 30. For each value of N, we obtain 1000 samples by bootstrapping. We plot N against the mean faithfulness score and include error bars for the standard deviation. For N >= 15, the mean faithfulness scores (i.e., Pearson correlation coefficients) are highly stable: they are all within $0.03$ of each other. We refer to this analysis In Section 6 in the main body of the paper and provide details in Appendix E.3. In addition, we now mention the small evaluation set size as a limitation in the paper (Section 6, lines 520-524).
> >
> > **Generalizability to open-source models: the paper carries analyses on two OpenAI models (GPT-3.5, GPT-4o) and one Anthropic model (Claude-3.5-sonnet). Could be interesting to contextualize the faithfulness of existing open source models (e.g., Llama 3.1) and draw comparisons with closed-source ones.**
> >
> > We agree with this point. We are working on adding experiments in which we assess the faithfulness of Llama 3.1 models. We will provide an update on that when we finish these new experiments. We expect this to be before the end of the review period.

---

> > > ### Author Response · Authors · 2024-11-22
> > > **Clarification of experiment details**
> > >
> > > We thank the reviewer for pointing out important missing details. We’ve now added them to the paper in Appendix D, which we refer to in the main body of the paper (see line 205). We state where each is below.
> > >
> > > To ensure full reproducibility, we will release our code with the final version of our paper. We plan to spend time in the near future documenting it carefully. However, if reviewers are interested in seeing it now, we can share it in a ZIP file.
> > >
> > > **Important parameters of the experiments are not clear in the paper, which may affect reproducibility of the experiments. The authors could consider providing additional details about the exact number of perturbations generated for each example, the number of generations used to estimate P(Y|X) (during the definition of CE), the decoding hyperparameters (including the temperature and max number of tokens). Additional details should also be provided about how each response y is parsed – this is particularly relevant given that the evaluated models are known to produce nuanced and semantically equivalent texts.**
> > >
> > > The details of our experimental settings are now included in the following places in the paper:
> > > * The exact number of perturbations generated for each example is in Table 17 (referred to in Appendix D.2).
> > > * The number of number of generations used to estimate P(Y|X) is in Appendix D.3.
> > > * The decoding parameters used for all auxiliary LLM steps are specified in Appendix D.1.
> > > * The decoding parameters used for the primary LLMs (i.e., those we measure the faithfulness of) are in Appendix D.3.
> > > * Details on the LLM response parsing for the auxiliary LLM steps are in Appendix D.1.
> > > * Details on response parsing for the primary LLMs (i.e., those we measure the faithfulness of) are in Appendix D.3.
> > >
> > > **The authors should also consider including the prompt used to extract concepts and concept values in the Appendix.**
> > >
> > > We now include the prompts used for the auxiliary LLM steps in Appendix D.1.
> > > * For concept extraction: we include the basic prompt template in Table 6, the prompt details for the social bias task in Table 7, and the prompt details for medical question answering in Table 8.
> > > * For concept values extraction: we include the basic prompt template in Table 9, the prompt details for the social bias task in Table 10, and the details for medical question answering in Table 11.

---

> > > > ### Author Response · Authors · 2024-11-22
> > > > **Auxiliary LLM: Model Choice and Outputs**
> > > >
> > > > We agree with the reviewer that several aspects regarding our use and validation of the auxiliary LLM were not clear in the original paper. Based on their feedback, we have (1) clarified our motivation for using GPT-4o as the LLM, (2) provided a random sample of example outputs (concepts, concept values, and counterfactuals), and (3) clarified our process for validating the quality of LLM outputs.
> > > >
> > > > **Q: One of the decisions in the paper is to use an auxiliary model to extract concepts, propose a list of alternative values for each concept. Why is this necessary and is there any assumption or requirement that helped settling for a GPT-4o as the auxiliary LLM? The authors could better motivate the selection of GPT-4o in their paper, perhaps by including a small human study comparing the effectiveness of different models in extracting concepts and creating the list of alternate values.**
> > > >
> > > > We use an auxiliary LLM for these steps because they would be time-consuming to perform manually. Hence, automating this step is important for the utility and scalability of our method.
> > > >
> > > > We agree that our motivation for choosing GPT-4o was not clear in the original version of the paper. We’ve now added a sentence motivating this choice (see Section 4.1 lines 280-282). We chose to use a GPT-based model as the auxiliary LLM because they have been used for this purpose in prior work. [1][2] use GPT-based models for counterfactual generation and find that they produce high-quality counterfactuals. We chose GPT-4o specifically because it is a state-of-the-art model.
> > > >
> > > > We like the idea of including a small human study. However, given the time, IRB approval, and financial costs entailed, we do not anticipate being able to execute this in time for the rebuttal. We would like to pursue it in future work.
> > > >
> > > > [1] Gat, Yair, et al. "Faithful explanations of black-box nlp models using llm-generated counterfactuals." arXiv preprint arXiv:2310.00603 (2023).
> > > > [2] Polyjuice: Generating Counterfactuals for Explaining, Evaluating, and Improving Models (Wu et al., ACL-IJCNLP 2021)
> > > >
> > > > **Q: In line 218, the authors mention the use of auxiliary LLM to “list distinct concepts in the context of x”. What kind of verifications were performed to ensure that the extracted concepts and their list of concepts were meaningful? The authors should consider adding a list of extracted concepts and their list of values to the Appendix. They should also consider adding more details about the validation (e.g., manual validation or llm-as-a-judge approach).**
> > > >
> > > > We manually validated the extracted concepts and concept values to confirm that they are plausible, meaningful, and distinct. For BBQ, we reviewed all concepts and concept values (of which there are 134 total). For MedQA, we reviewed concepts and concept values for a random sample of 15 out of the 30 questions (161 concept/concept values total). We have now included a random sample of the concepts and concept values in Appendix D.2. The concepts/values for BBQ are in Table 18 and the concepts/values for MedQA are in Table 21.
> > > >
> > > > **Q: Similarly, to the two questions above, in line 224-225, the authors mention “to generate each counterfactual, we instruct [auxiliary LLM] to edit the question x by changing the value of [concept] ..., while keeping everything else the same”. However there seems to be no validation of this claim. Did the authors validate that the perturbed input x was indeed minimally distant from x? If not, the authors should consider including such analysis, perhaps by showing the minimum edit distance or n gram overlap between the modified and original inputs.**
> > > >
> > > > We manually reviewed the counterfactuals to check that (1) the requested edit was made and (2) no other information in the question was altered. For BBQ, we reviewed all counterfactuals (268 total). For MedQA, we reviewed the counterfactuals for a random sample of 15 out of the 30 questions (161 counterfactuals total). We have now included a random sample of counterfactuals in Appendix D.2. For BBQ, the removal-based counterfactuals are in Table 19 and the replacement-based counterfactuals are in Table 20. For MedQA, all counterfactuals are removal-based and they are in Tables 22 and 23.

---

> > > > > ### Author Response · Authors · 2024-11-22
> > > > > **Other Comments**
> > > > >
> > > > > **Univariate counterfactuals: if I understood correctly, the proposed framework focuses on perturbing the sentences one concept at a time, irrespective of the correlations between features. However, this fails to account for the correlations between different features (e.g., name of schools or organizations is related to socio-demographic features).**
> > > > >
> > > > > The reviewer is correct in their understanding that we focus on single concept interventions, and that this can result in issues in the case of correlated concepts. We now mention this in the limitations section (Section 6 lines 528-530) and provide a more detailed discussion in Appendix G. In future work, we plan to examine multi-concept interventions as a way of addressing this.
> > > > >
> > > > > We would like to make a clarifying comment about the challenge of correlated concepts. As explained in Appendix G, correlated concepts are primarily an issue when generating counterfactuals that involve *removing* a concept. If multiple concepts are correlated in the data used to train an LLM (e.g., an individual's race and an individual's name), then even when a single concept (e.g., race) is removed from the input question, an LLM may still infer it using the information provided by the other concepts (e.g., name). However, generating counterfactuals that involve *replacing* the value of a concept (e.g., changing an individual's race from Black to White) can help to resolve this issue. This is because in this case, the LLM can use the provided value (e.g., White) of the concept intervened on rather than inferring it based on the other concepts.
> > > > >
> > > > > **Q: Why did the authors select a linear correlation coefficient as opposed to a non-linear coefficient?**
> > > > >
> > > > > Our goal is to assess the alignment between the causal effects of concepts and the rate at which they are mentioned in the LLM’s explanations. Intuitively, if an LLM’s explanations are faithful, then the “alignment” should be high. There are multiple metrics we could use to measure “alignment,” and it is not clear which is best. Since we are the first work to perform a faithfulness analysis of this kind, we opt to use a linear coefficient for simplicity. However, we think it would be interesting to examine multiple alignment metrics in future work.
> > > > >
> > > > > **Q: In Figure 1 (right), we observe that different behavioral concepts end in different regions of the scatter plot (there are orange points in the top and orange points around EE in [-0.5, -1.5]. Is there any insight or pattern that justify why there are different clusters? Could it be that the model is less prone to use some concepts for specific demographics?**
> > > > >
> > > > > As you observed, whereas the explanations given by GPT models consistently mention the behavior-related concepts (shown in orange), it appears that Claude sometimes mentions them (i.e., the high EE value cluster) and sometimes doesn’t (i.e., the low EE value cluster). When examining a subset of the Claude explanations manually, we found that they appear to follow one of two patterns: (1) they mention the behavior-related concepts as the reason for the decision or (2) they choose the “undetermined” answer choice and say that this is because it is not safe/ethical to answer the question (note: in this case, behavior concepts are not mentioned). It seems that pattern (1) corresponds to the first cluster you mentioned and pattern (2) corresponds to the second. We plan to add a discussion of this in the paper (after finding places to cut to make space for this).

---

> > > > > > ### Author Response · Authors · 2024-11-22
> > > > > > **Writing clarify suggestions**
> > > > > >
> > > > > > **Add information about when the experiments were run with each model.**
> > > > > >
> > > > > > Can you please clarify what you mean by this? Are you asking us to specify the dates on which we conducted our experiments? In the paper, we’ve included the specific releases of the LLM APIs that we use (e.g., gpt-4o-2024-05-13). Does this address your concern?
> > > > > >
> > > > > > **lines 321-323: you describe the colors for each of the concept categories. However there seems to be a mismatch between the category color in the image and the color described in text.**
> > > > > >
> > > > > > Thank you for catching this. We have now fixed it in the paper.

---

> > > > > > > ### Comment · Reviewer_7q6M · 2024-11-24
> > > > > > >
> > > > > > > Thank you for all your effort and time in answering my review. The authors addressed all my comments in a satisfactory way, so I'll revised my score accordingly

---

> ### Author Response · Authors · 2024-11-27
> **update on analysis of open source LLMs**
>
> We thank the reviewer for the time spent reviewing our rebuttal. We are happy that they found that we addressed their comments in a satisfactory way, and we greatly appreciate that they updated their score.
>
> As an update, we have now completed experiments with Llama-3.1-8B-Instruct as the LLM. We are still working on experiments with Llama-3.1-70B-Instruct – it has taken time to figure out how to run this model with our limited computational resources. We now have experiments running for the 70B-parameter model, and we will provide an update once they are complete.
>
> We plan to add the results from both of the Llama models to the main body of the paper once these experiments have finished. For now, we have included the results for Llama-3.1-8B-Instruct on the social bias task in Appendix E.5. Interestingly, we found that Llama-3.1-8B-Instruct obtains the highest faithfulness score (0.81) of all LLMs on the social bias task (GPT-3.5 obtains the second highest score of 0.75). This result further supports our finding that the smaller, less capable LLMs obtain higher faithfulness scores than the state-of-the-art LLMs. When we examined the concept category results, we found that Llama-3.1-8B exhibits the same pattern of unfaithfulness as the GPT models (although to a lesser degree): its explanations cite behavior-related concepts regardless of their causal effects and omit identity-related concepts regardless of their causal effects.
>
> We also applied Llama-3.1-8B-Instruct to the medical question answering task, using the same few-shot prompt we used for the other LLMs. However, we found that rather than answering the questions, Llama-3.1-8B-Instruct generates new questions. This makes our faithfulness analysis inapplicable (since there are no answers/explanations to analyze). So far, it appears that Llama-3.1-70B-Instruct does not have this same issue -- it answers the questions as expected. We plan to include results for this larger model on medical question answering.

---

### Official Review · Reviewer_L2o3 · 2024-11-12

**Soundness:** 3
**Presentation:** 4
**Contribution:** 3
**Rating:** 6
**Confidence:** 4

**Summary:**

This paper aims to measure the faithfulness of LLM generated natural language explanations in terms of specific concepts mentioned in it. Specifically, faithfulness is measured by the correlation between the causal effect of a concept (measured by counterfactual predictions) and the likelihood of the concept being mentioned in the explanation. This analysis produces several interesting results, e.g., the model doesn’t mention gender in its explanation despite gender having large causal effect on its prediction; safety alignment can affect model explanation.

**Strengths:**

- Inspecting more finegrained faithfulness is a novel contribution and it allows us to gain better understanding of specific biases in model explanations.
- The paper proposes a principled method to quantify faithfulness based on counterfactual examples.
- The finding that safety alignment can make the model hide true reasons (e.g., gender bias) in its explanation (thus is only a form of shallow alignment) is very interesting.

**Weaknesses:**

- It is unclear how the explanations are generated, e.g., are these CoT from zero shot or few shot prompting? Is explanation generated before or after model prediction? It would be interesting to analyze how different prompting methods change the result or improve faithfulness.

Minor:
- 152: distinct -> disentangled might be a more precise word
- 194: typo: x in the equation should be in vector form

**Questions:**

- Here the causal concept effect is considered the ground truth in some sense. Then would it make sense to directly explain the model prediction using the causal concept effect?
- For the dataset level faithfulness, instead of averaging question level faithfulness, why not directly measure PCC of all examples in the dataset?

---

> ### Author Response · Authors · 2024-11-22
>
> We thank the reviewer for their careful reading and valuable feedback. We appreciate that they recognize the novelty of our contribution, and in particular, our introduction of a principled method for assessing faithfulness in a fine-grained manner. We are grateful that they found the new insights about LLM faithfulness and safety alignment produced by our method interesting.
>
> We also thank the reviewer for helping to identify places where the clarity of our paper could be improved. We have revised the paper based on this feedback. We address individual points below.
>
> **It is unclear how the explanations are generated, e.g., are these CoT from zero shot or few shot prompting? Is explanation generated before or after model prediction? It would be interesting to analyze how different prompting methods change the result or improve faithfulness.**
>
> We agree that this was not clear in the original version of the paper. We have now clarified this in the main text (see Section 4.1 line 286 and Section 4.2 line 431) and provided the exact prompts used in Appendix D.3 (see Table 24 for the BBQ prompt and Table 25 for the MedQA prompt). In both the BBQ and MedQA experiments, we use a prompt that includes three few-shot examples and a chain-of-thought trigger (i.e., “let’s think step-by-step”). The prompt directs the model to produce the explanation before the prediction.
>
> We agree that it would be interesting to analyze the impact of the choice of prompting strategy on faithfulness. We are now working on an analysis in which we repeat our experiments on the social bias task using a different prompt. More specifically, we seek to understand if using a prompt that specifically directs the LLM to avoid social bias will lead to improved faithfulness. In our initial results, it appears that the difference is minimal, but we will provide a more complete report once this analysis is finished. We expect this to be done before the end of the review period.
>
> **Minor: 152: distinct -> disentangled might be a more precise word**
>
> We have updated the paper accordingly.
>
> **194: typo: x in the equation should be in vector form**
>
> Thank you for catching this. We have now updated it in the paper.

---

> > ### Author Response · Authors · 2024-11-22
> >
> > We thank the reviewer for their thoughtful questions. We respond to each below.
> >
> > **Q: Here the causal concept effect is considered the ground truth in some sense. Then would it make sense to directly explain the model prediction using the causal concept effect?**
> >
> > Yes, we think that our method for assessing the causal effects of concepts could itself be used as an explainability method. As we saw in our experiments in this paper, examining the magnitude of the causal effect of each concept provides an understanding of which factors influence LLM decisions and which do not. Existing work has explored the idea of using causal concept effects to explain model decisions in settings that are different from ours [1][2]. In particular, prior work assumes that all questions contain the same set of concepts and that concepts/concept values are manually annotated. In contrast, we allow questions to have different concepts and automatically extract them with an LLM. Moreover, we are the first work to present a Bayesian hierarchical modeling approach to estimating concept effects. This approach has the advantage of leveraging shared information across questions, while still allowing for questions to have different concepts and still capturing question-specific variability.
> >
> > While we think our method has promise as an explainability approach, additional work is needed to validate it for this purpose (e.g., experimental comparisons to [1][2]). That is out-of-scope for this paper, which is focused on examining LLM faithfulness. However, we plan to explore it in future work.
> >
> > [1] Gat, Yair, et al. "Faithful explanations of black-box nlp models using llm-generated counterfactuals." arXiv preprint arXiv:2310.00603 (2023).
> > [2] Abraham, Eldar D., et al. "Cebab: Estimating the causal effects of real-world concepts on nlp model behavior." Advances in Neural Information Processing Systems 35 (2022): 17582-17596.
> >
> > **Q: For the dataset level faithfulness, instead of averaging question level faithfulness, why not directly measure PCC of all examples in the dataset?**
> >
> > This is a good question. We thought about this carefully when developing our method.  The primary reason for averaging question-level faithfulness is that we think that measuring the dataset-level PCC can be misleading in some cases. In particular, it is possible to have a case in which an LLM’s explanations incorrectly order concepts by their causal effects *within* each question, but when looking *across* questions, the PCC is high (as in Simpon’s Paradox). This can happen if on certain questions the causal effects and explanation implied effects of concepts are systematically higher than on other questions. In this case, the low within-question PCC implies that the explanations provided for each individual question do not correctly refer to the most influential concepts for that question, which makes them unfaithful/misleading. But the high dataset-level PCC fails to capture this. We have now added a discussion of this point to Appendix G, which we refer to in the main body of the paper (line 201).

---

> > ### Author Response · Authors · 2024-11-27
> > **update on analysis of impact of prompting strategy on LLM faithfulness**
> >
> > As an update, we have now completed an analysis of the choice of prompting strategy on faithfulness. We have posted an updated version of the paper that includes this new experiment. We refer to it in the main body (see lines 414-417) and provide details in Appendix E.4.
> >
> > We repeated our experiments on the social bias task using a prompt that explicitly encourages the model to avoid stereotypes. Specifically, we use the same few-shot chain-of-thought prompt used in our other experiments, with one additional statement: “Please ensure that your answer is unbiased and does not rely on stereotypes.” For GPT-4o and Claude-3.5-Sonnet, we found that this choice of prompt had little effect on the results. For these two LLMs, the faithfulness scores changed by less than $0.04$ PCC, and the category-specific trends are mainly unchanged. For GPT-3.5, we found that using this “anti-bias” prompt *decreases* faithfulness. While this might appear somewhat surprising, our method surfaces concept-level unfaithfulness patterns that help to explain this. When using the anti-bias prompt, interventions on behavior-related concepts have less of a causal effect on GPT-3.5’s answers compared to using the standard prompt. Despite this, GPT-3.5’s explanations still reference behavior concepts at high rates. As a result, the prompt leads to a reduced faithfulness score. To further explain this, we observe that the reduced effects of the behavior concepts seems to stem from the fact that GPT-3.5 tends to more frequently select “undetermined” as its answer when using the anti-bias prompt, regardless of the intervention. Further details of this experiment are in Appendix E.4.
> >
> > We think this experiment nicely highlights the utility of our method. Our approach enables us to not only quantitatively assess the impact of the choice of prompting strategy on faithfulness but also to understand *why* using a certain prompt results in more or less faithful explanations. We thank the reviewer for suggesting this experiment, as we think including it substantially improved the paper.

---

### Author Response · Authors · 2024-11-22
**Response to Reviewer Feedback**

We thank all of the reviewers for taking the time to carefully review our paper and to provide thoughtful feedback and questions. Their feedback has helped us to strengthen the paper.

We have now uploaded a new version in which we have made changes based on reviewer feedback. In the main body of the paper, we have indicated the parts we have added in red. We have also added multiple new appendices.

We respond to the comments of each individual reviewer by replying to them directly below.

---

### Meta-Review · Area_Chair_XDBm · 2024-12-20

**Metareview:**

This paper studies the question of whether explanations provided by LLMs for their behaviors are in fact faithful to the actual behaviors. This requires formalizing faithfulness, developing a methdology to measure it, and then validating the methodology. All reviewers appreciated the problem, found the paper clear, and the contribution significant. The main concern here was whether the method was sufficiently validated, but overall this seems to be a minor consideration.

**Additional Comments On Reviewer Discussion:**

All reviewers were positive at all stages

---

### Decision · Program_Chairs · 2025-01-22

Accept (Spotlight)